# Functional Nanofibrous Biomaterials of Tailored Structures for Drug Delivery—A Critical Review

**DOI:** 10.3390/pharmaceutics12060522

**Published:** 2020-06-08

**Authors:** Zhen Li, Shunqi Mei, Yajie Dong, Fenghua She, Yongzhen Li, Puwang Li, Lingxue Kong

**Affiliations:** 1Institute for Frontier Materials, Deakin University, Geelong, Victoria 3216, Australia; zkj@deakin.edu.au (Z.L.); dongy@deakin.edu.au (Y.D.); mary.she@deakin.edu.au (F.S.); 2School of Mechanical Engineering and Automation, Wuhan Textile University, Wuhan 430073, China; 3Hubei Key Laboratory of Digital Textile Equipment, Wuhan Textile University, Wuhan 430073, China; 4Key laboratory of Tropical Crop Products Processing, Ministry of Agriculture and Rural Affairs, Agriculture Products Processing Research Institute, CATAS, Zhanjiang 524001, China; liyongzhen@catas.cn (Y.L.); puwangli@yahoo.com (P.L.)

**Keywords:** nanofibrous biomaterials, nature fiber biomaterials, biopolymers, drug delivery, nanofiber technology

## Abstract

Nanofibrous biomaterials have huge potential for drug delivery, due to their structural features and functions that are similar to the native extracellular matrix (ECM). A wide range of natural and polymeric materials can be employed to produce nanofibrous biomaterials. This review introduces the major natural and synthetic biomaterials for production of nanofibers that are biocompatible and biodegradable. Different technologies and their corresponding advantages and disadvantages for manufacturing nanofibrous biomaterials for drug delivery were also reported. The morphologies and structures of nanofibers can be tailor-designed and processed by carefully selecting suitable biomaterials and fabrication methods, while the functionality of nanofibrous biomaterials can be improved by modifying the surface. The loading and releasing of drug molecules, which play a significant role in the effectiveness of drug delivery, are also surveyed. This review provides insight into the fabrication of functional polymeric nanofibers for drug delivery.

## 1. Introduction

Nanofibers are a significant kind of biomaterial that could be used for biomedical applications, due to their special structure and properties such as high surface area [1], superior mechanical properties [2], high porosity [3], and low density [4]. Drug delivery is one of the most important emerging applications of nanofibers [5,6], because nanofibers have similar structural features and functions to those of extracellular matrix (ECM). The ideal drug delivery system can deliver and release a well-controlled amount of drug for a suitable period of time into a target site of the human body [7].

Nanofibrous biomaterials can be prepared from a wide range of polymers for drug delivery [8]. Polymeric biomaterials can be divided into natural and synthetic polymeric biomaterials. Natural polymeric biomaterials include chitosan, chitin, cellulose, gelatin, collagen, pectin, proteins, gelatin, and lignin [9]. These natural polymers are biocompatible and can be used to mimic ECM [10]. However, they are very difficult to form into continuous nanofibers [5]. Therefore, synthetic polymeric biomaterials with biodegradable properties have been composited with those natural polymeric biomaterials, due to their molecular weights being long enough to fabricate continuous nanofibers after elongation. Polymers that have been approved as biomaterials such as polyethylene oxide (PEO), polycaprolactone (PCL), poly(lactic-*co*-glycolic) acid (PLGA) and Poly(*N*-vinylpyrrolidone) (PVP) are usually utilized to form composites with natural polymers for nanofiber fabrication and for sustainable and controlled drug release [11].

Due to the superior properties of nanofibers, various nanofiber production technologies have been studied and utilized by many studies, including electrospinning, centrifugal spinning, solution blowing, phase separation, and self-assembly. Recently, electrospinning has been one of the major methods for nanofiber production, because of its numerous advantages, such as simple principles and equipment, broad material choice, and fabrication of nanofibers with versatile and uniform morphologies [12,13,14]. Other technologies for nanofiber production have also been reported and studied by many researchers [15]. The advantages and disadvantages of those technologies for fabrication of functional nanofiber scaffolds for drug delivery are reported.

Morphology and structure of nanofibrous biomaterials also significantly influence the function and effectiveness of drug delivery [16]. The morphology and structure involve fiber diameter, fiber cross-section shape, directionality, porosity and dimensionality of scaffold. For example, natural ECMs are usually highly 3D porous collagen nanofibers with diameters in the range of 50–500 nm [11]. In addition, many tissues (like tendon, muscle tissues, ligament and tympanic), cells and ECMs are highly aligned. Therefore, the fabricated nanofiber scaffolds should have similar morphology and structure to mimic the native ECM during delivery of drugs and regenerate damaged tissue.

The drug loading methods and drug release rate significantly influence the effect of drug delivery. Drug loading methods can be divided into chemical and physical adsorptions. Drug release rate from nanofibers is determined by various factors, including drug diffusion, fiber erosion and biodegradation. This review introduces the current state of development in the field of drug loading molecules on nanofibers for drug delivery. It will be followed by discussion and comparison of various nanofiber production technologies. The current challenges and perspectives of nanofiber scaffolds for drug delivery are presented, and the future research directions of the field are also highlighted.

## 2. Variety of Polymeric Biomaterials

Over 200 polymers can be utilized to spin nanofibers; however, only those that are biocompatible and biodegradable have been utilized as biomaterials to load drugs for tissue engineering [17]. Table 1 presents various biocompatible and biodegradable polymers that have been used to produce nanofibers for different biomedical applications. Cellulose, chitosan, chitin and collagen are the major nature biopolymers; poly lactic-co-glycolic acid (PLGA), polyethylene oxide (PEO) and polycaprolactone (PCL) are popular synthetic biopolymers. Natural and synthetic polymeric biomaterials are usually composited to produce nanofiber scaffolds for various biomedical applications, as shown in Figure 1. Natural polymeric biomaterials are native extracellular matrixes (ECMs); however, they are very difficult to form into continuous nanofibers. Synthetic polymeric biomaterials are used to improve the spinnability and dimensional stability of nanofibers. In addition, the biodegradation rate of nanofibers also can be controlled by varying the ratio of nature biopolymers and synthetic biopolymers, so as to control the drug release rate during drug delivery.

### 2.1. Natural Polymeric Biomaterials

#### 2.1.1. Cellulose

Cellulose is a popular polysaccharide and exists throughout the world. Due to their excellent biodegradability and chemical resistance, cellulose nanofibers show a great potential for biomedical applications [46,47]. The process of cellulose is limited by the solubility of cellulose in common organic solutions, due to vast intermolecular and intramolecular hydrogen bonds [48]. Therefore, cellulose has to be dissolved into single or mixed solvents before producing nanofibers. Various solvents have been used to dissolve cellulose, including ethanol, water, chloroform, dioxin, *N*,*N*-dimethylacetamide (DMA), dimethylformamide (DMF) and dichloromethane (DCM) (cellulose is quite difficult to dissolve) [46,49,50,51]. Initially, cellulose fibers were produced by wet spinning, but, currently, cellulose nanofibers are usually fabricated via electrospinning [46]. Cellulose derivatives are also widely used to deliver drugs or growth factors in tissue engineering applications. Cellulose acetate (CA) is obtained from acetylating cellulose, the most abundant natural polymer [47]. CA has huge potential for drug delivery. CA fiber mats produced via electrospinning were utilized to release three ester prodrugs of naproxen [52] and to load tetracycline hydrochloride and slowly release the drug for antibacterial activity [48]. CA nanofibers also showed their antioxidant characteristics via loading 6-gingerol for transdermal drug delivery [53]. Ethyl cellulose (EC) was used as the shell layer of core–shell nanofibers to protect the core layer and release bioactive agents during in-vitro cell culture studies [54]. In addition, cellulose triacetate, methyl cellulose and hydroxypropyl cellulose were also investigated for tissue engineering applications [55,56,57].

#### 2.1.2. Chitin and Chitosan

Chitin is a linear 1, 4-linked polymer composed of *N*-acetyl-d-glucosamine residues, which can be obtained from seafood wastes and invertebrate skeletons [58]. Chitin is one of the largest abundant natural polysaccharide polymers in the world [59]. It is also one of the most promising natural polymers for tissue engineering applications, due to its biocompatible, biodegradable, antibacterial, nontoxic, and adhesive properties [44]. Chitin and its derivatives have been prepared to produce various forms of materials (including nanofibers, membranes and sponges) for wound dressing and burn dressing. Chitin-based dressings could accelerate contraction of wounds, promote repairing of damaged tissues, and regulate secretion of inflammatory mediators. Chitin nanofibers have been seen as an ideal substitute for traditional inorganic nanofillers for drug delivery applications, not only because of their biodegradability and biocompatibility, but also their excellent mechanical properties. However, chitin cannot be dissolved into common organic solvents or diluted aqueous, and the weak solubility of chitin hinders its industrialization [60]. Therefore, finding a suitable dissolution system for chitin is essential for further extending the applications of chitin.

Chitosan is a linear polymer composed of β (1-4) linked d-glucosamine units, which is derived from *N*-deacetylation of chitin. As a significant derivative of chitin, chitosan is also biodegradable, biocompatible and nontoxic. Moreover, due to its antibacterial and antifungal properties, chitosan has great potential for tissue engineering applications. Recently, chitosan-based nanofibers have been utilized as matrix molecules for drug delivery [61]. Their application in drug delivery is significantly influenced by the degree of acetylation and molecular weight because these properties influence hydrophobic ability and can change the drug encapsulation efficiency. although chitosan can hardly be dissolved in neutral aqueous solvents, its solubility improves with increasing acidic solvents because of its amino groups [62]. As a vehicle of drug delivery, the mucoadhesive ability of chitosan has attracted much attention. Chitosan has been used to load and deliver drugs through various epithelia, including buccal [63], ocular [64], intestinal [65], and nasal [66]. The spinnability of pure chitosan is poor; therefore, many synthetic polymers are used to composite with chitosan to produce chitosan-based nanofibers [67]. Chitosan cannot be dissolved into most organic solvents; therefore, various chitosan derivatives have been prepared to improve the encapsulation ability of hydrophobic drugs. For example, carboxylated chitosan (CCS) is used as a water-soluble chitosan to fabricate chitosan-based nanofibers for delivery of drugs in skin regeneration [68]. In addition, water-soluble chitosan could be used for wound healing applications, rather than being restricted by toxic or acidic solvents [69].

#### 2.1.3. Collagen

Collagen is not only the primary structural element of ECM, but also the most abundant protein of the human body [45]. Collagen is organized into insoluble fibers to support tensile strength. For example, muscle fibers transmit forces, consume energy and protect tissues from external forces, due to numerous collagens in them. If collagen is insufficient, the tissue is weak and might rupture [70]. Moreover, collagen also provides biological cues to nearby cells and regulates various bio-functional responses [20]. The collagen family involves at least 30 different gene products and concentrates into over 20 collagen types. The molecular structure of these collagen types is the triple helical. Additionally, collagen types I, II and III are the most abundant fibrillar collagens in the human body [11]. Collagen has been applied into a large number of tissue engineering applications, due to its excellent properties in the ECM, low antigenicity, non-immunogenicity and cell compatibility [71]. Collagen was used in orthopedic surgeries as an implantable ECM to accelerate bone growth [72]. Extracted collagen is quite difficult to process into artificial nanofibers. Therefore, researchers usually dissolve collagen with other spinnable polymers into solvents and produce collagen composite nanofibers for drug delivery in tissue engineering. Collagen type I was coated on PCL–chitosan nanofiber to bind fiber scaffolds and as an agent for healing burn injuries during skin regeneration [73]. Collagen and PLLA were blended into HFIP and produced nanofibers via electrospinning as wound dressing [74]. Three-dimensional (3D) PLGA nanofiber scaffolds loaded with collagen I were utilized to promote primary hepatocyte function [75]. A novel collagen-mimetic peptide amphiphile has also been produced to make collagen-based nanofibers for tissue regeneration [76].

#### 2.1.4. Other Natural Polymeric Biomaterials

Other natural polymeric biomaterials, such as silk fibroin, keratin, alginate, and chondroitin, are also broadly studied for drug delivery and tissue engineering. For example, silk fibroin is derived from cocoons, which is a promising biopolymer due to its excellent biocompatibility and low biodegradation rate in the human body [77]. Ang et al. produced silk fibroin composite nanofibers to deliver osteogenic marker genes, osteocalcin and alkaline phosphatase for bone tissue engineering [78]. For drug delivery applications, these biopolymers are frequently produced for implantation as porous nanofiber scaffolds or nanofiber membranes into nontoxic ending products in vivo.

However, the disadvantages of natural biopolymers include inconsistent compositions and weak mechanical strength [79]. Additionally, the kinetics of these natural biomaterials might be hard to control when the long-term responsive action is insufficient.

### 2.2. Synthetic Polymeric Biomaterials

#### 2.2.1. Poly Lactic-*co*-Glycolic Acid (PLGA)

Poly lactic-*co*-glycolic acid (PLGA) is a co-polymer material from poly lactic acid (PLA) and poly glycolic acid (PGA) with different ratios, such as 75:25, 65:35, 50:50 and 25:75 [11]. The melting point and crystallinity degree of polymers are ultimately related to their molecular weight. The glass transition temperature of PLGA has been demonstrated to decrease with decreasing lactide content and molecular weight [80]. The mechanical strength of PLGA nanofibers is significantly influenced by their physical properties, including polydispersity index, ratio of poly lactic acid and poly glycolic acid, and molecular weight. These properties also impact the shape and size of PLGA production for delivery of drug and controlling the degradation rate [81]. PLGA is one of the most attractive synthetic polymers, and is frequently employed to prepare materials for drug delivery, due to its biocompatible, biodegradable and tunable mechanical properties. PLGA has been widely investigated and processed into any morphology for development of biomedical materials for delivery, control and release of bioactive agents, drugs and proteins in the academic community and industry. PLGA has been combined with other materials, including bioactive glass or ceramics, to improve biomimetic ability and accelerate bone regeneration. Porous silica nanoparticles are random loaded into PLGA nanofibers via the electrospinning method for improving mechanical properties in cell proliferation [32]. PLGA is widely dissolved into many common solvents, such as HFIP [32,37], DMF [30,34], THF [31], chloroform [75] and ethyl acetate [35], for drug delivery in biomedical applications. However, the potential residual toxic solvents might pose a negative influence for drug release and cell proliferation.

#### 2.2.2. Polycaprolactone (PCL)

Polycaprolactone (PCL) has been widely explored due to its excellent properties (biocompatibility, biodegradation, non-toxicity, low melting point (60 °C) and semi-crystallinity) and low cost. In addition, PCL can be dissolved into many common solvents, such as HFIP, chloroform, acetic acid, methanol and dichloromethane [37,82,83]. Due to these advantages, PCL is frequently utilized to produce multi-functional nanofibers for drug delivery in tissue engineering applications. Some common solvents are usually mixed to combine PCL with hydrophilic drugs, because PCL is a hydrophobic biomaterial [8]. For example, PCL–gelatin composite nanofibers were used to load metronidazole for anti-infection of skin tissue regeneration [84]. Additionally, smooth, homogeneous and hydrophilic PCL–gelatin nanofibers were used to grow and proliferate human umbilical arterial smooth muscle cells. PCL has also been blended with chitosan to promote the biocompatible and hydrophilic properties of nanofibers to mimic ECM and guide cell proliferation [85,86].

#### 2.2.3. Polyethylene Oxide (PEO)

Polyethylene oxide (PEO) is a crystalline synthetic polymer with thermoplastic properties. It is a water-soluble polymer with a simple chemical formula, H-(OCH_2_CH_2_)_n_-OH. Compared with other water-soluble synthetic polymeric biomaterials, PEO is unique in its linear structure. This special linear structure represents an excellent polymer–solvent interaction in water. For nanofiber production, the molecular weight of PEO is usually between 300,000–7,000,000. PEO is a particularly effective synthetic polymer for protein resistance because of its hydrophilicity [87]. Moreover, PEO is frequently employed into drug delivery due to its biocompatibility, biodegradability, and non-toxicity [88]. Qu et al. produced PEO nanofibers to deliver the targeted enzyme for meniscus repair [89]. Additionally, PEO could be used to improve dimensional stability of nanofiber meshes [90]. Sadri et al. blended PEO with chitosan to enhance the spinnability of chitosan for antimicrobial agents in wound healing [91].

#### 2.2.4. Other Synthetic Polymeric Biomaterials

Other biocompatible and biodegradable synthetic polymers are also extensively studied for drug delivery in tissue engineering applications, such as PVA and PVP. Polyvinyl alcohol (PVA) is a water-soluble polymer and widely utilized by blending with other biopolymers. It has been combined with chitosan to enhance cell attachment and biocompatibility of the composite nanofibers [92]. Poly(*N*-vinylpyrrolidone) (PVP) is also a water-soluble polymer with low toxicity and chemical stability. PVP has been employed as a blood plasma substitute in drug delivery systems, due to its blood compatibility and physiological inactivity [93].

## 3. Nanofiber Production Methods

In order to develop a more effective and higher production rate method for fabrication of nanofibers, various kinds of nanofiber fabrication techniques have been researched, such as electrospinning, centrifugal spinning, airbrushing, wet spinning, simple freeze-drying and jet-rapid freezing. A great diversity of methodologies has encouraged a wide range of research on producing nanostructures with advantageous properties for a vast number of engineering applications [15,94,95].

### 3.1. Electrospinning

Electrospinning was used for fabricating continuous fibers, which was first patented and developed by Farmhals in 1934 [96]. In 1969, Taylor researched the shape of the droplet at the tip of the spinneret in the electric field, before the solution jet ejecting from the orifice of the spinneret [97]. Since then, the droplet is known as “Taylor cone”, as shown in Figure 2. Electrospinning is currently the most significant technology for manufacturing polymer nanofibers [98].

In electrospinning, a strong electric field (usually in the range of 5 to 30 kV) is generated between the polymer solution and a grounded collection plate by connecting the needle of the spinneret with a high voltage power, and connecting the collector with the ground (Figure 2). When a high voltage is supplied, the pendent drop of polymer liquid in the orifice of the spinneret becomes highly electrified and the induced charges are equally distributed on the surface of the spinneret, then the Taylor cone forms. Therefore, the liquid drop experiences electrostatic repulsion and Coulombic force at the same time [98]. There are various electrospinning parameters which can be divided into two categories: fluid intrinsic properties and operational conditions. The fluid intrinsic properties mainly include surface tension, solution viscosity, solution conductivity, molecular weight, and solvent evaporation rate. The operational conditions are voltage value, solution flow rate, nozzle diameter, collector distance, and spinning environment. The production rate of traditional electrospinning is too low to satisfy industrialization; therefore, various designs and equipment have been developed to improve productivity in the past decades.

Currently, the electrospinning system is mainly divided into two categories: needle electrospinning and needleless electrospinning. Needle electrospinning involves single-needle electrospinning (traditional electrospinning), multi-needle electrospinning and multiaxial electrospinning. The production rate of single-needle electrospinning is only around 0.1 mL/h; therefore, a straightforward method to improve the productivity is to increase the number of needles [99], and a waterfall geometry electrospinning setup with three needles was produced [100], as shown in Figure 3. However, in the multi-needle electrospinning system, strong electric field interference among the jets may reduce the production rate and form fibers of poor morphology and diameter distribution. Multiaxial electrospinning is designed to produce multiaxial nanofibers, even though the production rate is as low as traditional needle electrospinning, as shown in Figure 4. Various cross-section shapes of nanofibers can be produced via multiaxial electrospinning, to prevent bioactive agents from reaching the wound environment and load multiple drugs to improve the functionalization of nanofibers.

A needleless electrospinning system dramatically improves the productivity of the electrospinning method, compared with needle electrospinning. In a needleless electrospinning system, Taylor cones are created on the surface of the polymer solution which cover the fiber generator [101]. For this reason, the inter-molecular interactions in the solution have to be strong enough to stabilize these Taylor cones, to make sure that Taylor cones can be stretched into ideal jets and collected on the collector wall [102]. In the setup, polymer jets had been created on the surface of a positively charged rotating roller electrode which was half-immersed in a polymer solution reservoir. As a further development of the technology, the rotating roller has been replaced by a stationary wire electrode (Figure 5). However, needleless electrospinning has its drawbacks in guided tissue regeneration (GTR), as the needleless system cannot fabricate multiaxial nanofibers to protect bioactive agents and control drug release. Hence, the drug delivery abilities of needless electrospinning are lower than of multiaxial nanofiber electrospinning.

Electrospinning is the most popular nanotechnology. However, the low production rate (needle electrospinning) and high energy consumption limit the extensive range of commercial applications of nanofibers to take advantage of the unique properties. For these reasons, it is highly necessary to develop more effective approaches to produce nanofibers.

### 3.2. Centrifugal Spinning

Centrifugal spinning is an alternative technological method to fabricate polymer nanofibers at a high production rate but a low energy consumption, even though it is not a novel technology. This technique was evolved from a fabrication technology for cotton candy invented in 1897 [104]. Centrifugal spinning utilizes the centrifugal force to overcome the surface tension between the polymer solution and the nozzle wall, then the polymer jet is ejected and stretched by various forces (including centrifugal force, aerodynamic force, elastic viscous force and the Coriolis force), and the solvent continuously evaporates until the jet solidifies, and, finally, nanofibers are collected on a collector, as shown in Figure 6a. Similar to electrospinning, the parameters of centrifugal spinning can also be divided into two categories: fluid intrinsic properties and operational conditions. And these two nanofiber fabrication methods have some common parameters. The fluid intrinsic properties of centrifugal spinning are almost shared with electrospinning, except for solution conductivity. Therefore, the material choice of centrifugal spinning is broader than electrospinning because any polymer solution can be used to produce nanofibers via centrifugal spinning, even if the solution has no conductivity.

Centrifugal spinning is an emerging nanofiber fabrication method because it not only has a high production rate with a low cost, but can also produce multiaxial nanofibers to improve the functions of nanofibers [16,105]. Multiaxial structural nanofibers can be fabricated by different multiaxial nozzles, as shown in Figure 6b. This demonstrates that the centrifugal spinning method combines the advantages of needle electrospinning and needleless electrospinning. However, understanding the flow field of the technology and customizing the nanofiber structures for various applications are still to be further studied and investigated.

### 3.3. Solution Blowing

Solution blowing is a simple alternative method for nanofiber fabrication, due to the setup of this technology being simple, as shown in Figure 7. The solution blowing method is optimized from melt blowing [107]. In melt blowing system, polymer has to be melted at high temperatures, which significantly restricts its application in tissue engineering, due to the bioactive agents rapidly losing bioactivity in the environment. In order to overcome this disadvantage, polymer solution replaces melted polymer and the solution blowing method is introduced. Currently, non-woven nanofiber meshes for biomedical applications are frequently produced via solution blowing [108]. Singh et al. utilized solution blowing to fabricate core–shell PCL-PEO isotropic nanofibers for controlled sustainable release of dual drugs (bovine serum albumin and bFGF) [109]. However, how to produce aligned nanofibers in solution blowing system still needs to be solved, because numerous aligned artificial ECMs also have to be produced for respective applications.

### 3.4. Other Nanofiber Fabrication Techniques

Self-assembly is a process by which nanofibers are manufactured by holding molecules without external guidance or management. The technology of self-assembly can be divided into two types, intramolecular and intermolecular self-assembly, respectively [76,110]. In self-assembly approach, various mechanisms can be utilized to produce nanofibers depending on the polymer chemical structures. The mechanism of producing hydrogel is widely utilized to form network structural nanofibers via self-assembly of hydrogelator molecules.

Phase separation is another technique to fabricate polymeric nanofibers. The process of phase separation includes polymer dissolution, gelation, phase separation, solvent removal, and drying [26]. First of all, the polymer material is dissolved into a solvent, so as to form a homogeneous polymer solution. Then, the solution is sustained at the gelation time for several hours, so as to become nanofibrous matrix. Finally, nanofibers will be formed after evaporation of the solvent. It is clear that this kind of nanotechnology cannot produce aligned and multiaxial fibers; besides, the uniformity of nanofiber diameters cannot be guaranteed.

## 4. Morphologies of Nanofibers

The native ECM has a dynamic and 3D porous structure with a diameter at the range of 50–500 nm. Therefore, the morphologies of artificial nanofibers for tissue engineering and drug delivery should be similar to native ECM, so as to provide an ideal microenvironment for cell adhesion and proliferation. The morphologies of nanofibers mainly involve fiber orientation and cross-section shape.

### 4.1. Fiber Orientation

The fiber orientation can be divided into isotropic and anisotropic fibers, as shown in Figure 8. Different orientation morphologies can be used to mimic different native ECMs for specific tissue regeneration. The orientations of isotropic nanofibers (or unaligned nanofibers) are chaotic and random, which means that the mechanical properties of this nanofiber are uniform in all orientations. This morphology of nanofibers can be employed to repair skin tissue in wound healing, as the ECM fibers of skin are also unaligned. For example, isotropic PCL nanofibers have been used to adsorb chitosan nanoparticles containing human granulocyte colony stimulating factor (G-CSF) for skin tissue regeneration [111]. Additionally, heparin mimetic peptide nanofibers with unaligned morphology dramatically promoted the tissue regeneration of burn injury [112]. Besides, anisotropic nanofibers imply different properties in different orientations. Anisotropic nanofibers also have a broad application in tissue engineering because they can be used to mimic aligned native ECM, such as muscle and nerve fibers. Aligned nanofibrous scaffolds possess unique electrical, optical, and mechanical properties and are excellent materials to guide cell growth with the desired anisotropy [17]. The mechanical properties of aligned collagen–PCL nanofiber scaffolds were similar to heart valve leaflet and cardiac muscle [113]. Zhang et. al fabricated gelatin/PLLA nanofibrous scaffolds and demonstrated that the anisotropic nanofibrous scaffolds were very valuable in corneal tissue engineering [114].

### 4.2. Fiber Cross-Section

The cross-sectional shapes of nanofibers are various, including single nanofibers and multiaxial nanofibers (such as coaxial nanofibers, hollow nanofibers and islands-in-the-sea nanofibers), as shown in Figure 9. Single nanofibers are the most common nanofibers for drug delivery in tissue engineering. As reported in Section 3, simple physical adsorption, nanoparticle assembly and chemical adsorption of drugs and bioactive agents are usually loaded on the surface of single nanofibers because the production process of single nanofibers is much easier than for other structural nanofibers. However, the disadvantages of single nanofibers are also obvious. For example, bioactive agents on the surface of single nanofibers will lose bioactivity rapidly in wound healing applications, due to the initial wound environment being severe for bioactive agents. In addition, it is very difficult to control the release rate of drugs or bioactive agents, when they are simply adsorbed on the surface of single nanofibers.

Multiaxial nanofibers are able to encapsulate the drug into the nanofiber core, so as to provide protection from the surrounding environment and control drug release [116]. Many common polymers (such as cellulose, chitosan, PVA and PEO) can be used to produce multiaxial nanofibers, and various drugs (including growth factors, DNA, antibodies and proteins) have been loaded into multiaxial nanofibers with different layers for different purposes [117,118,119]. Successful drug encapsulation is dependent on accurately distributing the drug into the fiber core. Drug encapsulation efficiency is significantly influenced by drug properties (like stability and solubility) and nanofiber morphologies [120]. Several nanofiber production methods have successfully been used to fabricate multiaxial nanofibers for drug delivery in tissue engineering, such as electrospinning [121], airbrush [109] and centrifugal spinning [105]. Multilayer nanofibers can load multi-drugs to satisfy multifunction of the nanofibers for tissue engineering. Core–shell nanofibers provided dual drug release profiles with adjustable doses in the second phase of tissue regeneration [122]. In order to investigate the potential advantages of multilayer nanofibers, other types of structures have also been produced, including triaxial structural nanofibers, hollow structural nanofibers and islands-in-the-sea nanofibers. Triaxial structural nanofibers have provided various drug release profiles for different model drugs (Keyacid uranine and Keyacid blue) separately loaded into the shell layer and core layer of the fiber, with a PCL intermediate layer for slowing release of the drug in the core layer [116]. Resveratrol and gentamycin sulfate have been encapsulated by PCL hollow nanofibers and exhibited a sustainable release without drug bursting [123]. Islands-in-the-sea (multichannel structural) nanofibers might have unique advantages, including independent channels for individual drugs and better mechanical stability, for vessel devices and multi-drug delivery in tissue engineering applications [124].

## 5. Drug Loading in Nanofibers

### 5.1. Chemical Adsorption

Chemical adsorption: drugs and bioactive agents are modified onto the surface of nanofibers through various functional groups, including carboxyl groups, amine groups, hydroxyl groups and hydrophilic linkers, as shown in Figure 10. In fact, the chemical adsorption method is more favored than the physical adsorption method in biomedical applications, due to the bioactive agents being covalently immobilized to nanofibers [127]. Therefore, these bioactive agents are not easily split away from the nanofibers during the incubation period. However, it is notable that the modified agents will incur partial inactivation upon the chemically modified covalent positions.

Carboxyl groups have been frequently used to immobilize bioactive agents onto the surface of nanofibers. Carboxyl groups, for example, were successfully grafted onto the surface of PLGA, PLLA and PGA nanofiber scaffolds for conjugation with collagen, to improve cell adhesion and proliferation [128]. Carboxylic acid groups were employed to load rosuvastatin and heparin on the surface of cellulose acetate nanofibers for endovascular procedures [129]. Amine groups are also extensively used for covalent modification of polymeric nanofibers due to their high reactivity. Epidermal growth factor (EGF) has been chemically modified onto the surface of polymeric composite nanofibers for promoting the effect on the wound healing process [130]. Gold nanoparticles–SBA-15 composite has been bonded with Schiff bases via amine groups for improving the stability of biomaterials [131]. Hydroxyl groups are another kind of significant functional group, which also have been widely employed for chemical adsorption of drugs and bioactive agents. Hydroxyl groups of mesoporous silica nanoparticles (MSNPs) strongly interact with the nitrogenous groups of gelatin and the functional groups of PLGA to form a strong intermolecular network between those biomaterials for cell attachment and proliferation in wound healing [32,33].

However, direct modification of drug molecules on the surface of nanofibers might exhibit some limitations for cell attachment and proliferation. For example, cells cannot easily recognize the modified bioactive agents, as these agents are not entirely exposed to cells. Therefore, hydrophilic linkers are introduced to combine bioactive agents and nanofibers for promotion of cellular recognition. For example, EGF has been chemically modified onto the surface of amine-terminated block polymer composite nanofibers via polyethylene glycol (PEG) linkers [132].

### 5.2. Physical Adsorption

Currently, there are mainly three methods for physical adsorption of drugs in nanofibers or on the surface of nanofibers: simple physical adsorption, nanoparticle assembly, and multilayer assembly, as shown in Figure 11.

Simple physical adsorption: nanofibers have an extremely high porosity and surface area, allowing for a great number of drugs to be adsorbed on the surface of nanofibers. This is the simplest method for the delivery of drug via nanofibers, which can be used for burst release of drugs from the surface of nanofibers. PCL-gum tragacanth–curcumin nanofibers have been produced via dissolving curcumin into PCL-gum tragacanth solution, for preventing bacterial infection within a few hours during wound healing [133]. Heparin was successfully loaded on the surface of PEO/PLGA composite nanofibers by electrospinning [134]. In addition, anti-adhesion barrier application also requires loading drugs on the nanofiber surface, due to the adhesion between internal tissues often occurring after surgery [135].

Nanoparticles assembly: drugs are encapsulated into polymer nanoparticles, before these nanoparticles are adsorbed on the nanofibers. This method can effectively promote the drug loading capacity [136]. PVP nanofiber meshes containing silver nanoparticles have improved antibacterial properties in wound dressing [137]. Moreover, combining nanofibers and polymeric nanoparticles can enhance the multi-functional performance of nanoparticle-on-nanofiber structures [138]. Vascular endothelial growth factor (VEGF) has been relatively rapidly released on the surface of chitosan–PEO nanofibers, while platelet-derived growth factor-BB (PDGF-BB) is sustainably released from PLGA nanoparticles within the nanofibers [23]. A nanoparticle-on-nanofiber structure is a relatively stable system, which provides a one-step surface modification approach for drug loading.

Multilayer assembly: coaxial and triaxial micro–nano fibers with unique features (including reinforced core and hollow structure) can be utilized to sequester stimulants in different compartments and control the release of various drugs through changing drug positions and thickness of layers [8]. This drug delivery method has great potential for biomedical applications due to the generalization of any complex structure and the possibility of utilizing any composition for the core layer. For example, plasmid DNA has been embedded into the core layer of core–shell nanofibers for preventing enzyme attack before being released [139]. In addition, many bioactive agents have also been loaded into multilayer nanofibers. For instance, bovine serum albumin and basic fibroblast growth factor (bFGF) enjoyed sustained release from PCL-PEO composite core–shell structural nanofibers [109]. Notably, the thickness of multilayer fibrous structure also impact the drug release profile.

## 6. Drug Release from Nanofibers

Drugs or bioactive molecules are transported through the drug delivery system, which is controlled by the random movement of drug molecules and actuated by chemical potential gradients. A classical drug release in nanofibers refers to the drug being transported from its initial position in nanofibers to the outer surface of the nanofibers; then, the drug is released into its surroundings [140]. Additionally, drugs could be released through nanofibrous biomaterials by the matrix erosion effect, which results in porous formation. This section briefly discusses the selected examples of drug release mechanisms based on nanofibrous biomaterials. The mechanisms of drug release in nanofibers depend on drug diffusion, polymer nanofiber biodegradation, and nanofiber erosion [141].

### 6.1. Drug Diffusion

Drug molecules could be transported from nanofibers via diffusion through pores of nanofibers. It is known that nanofibers are highly porous; when nanofibers are filled with liquid, drug molecules randomly move through liquid-filled pores due to them being driven by the chemical potential gradients [142]. As time goes on, the size of pores becomes larger and more pores appear due to the degradable properties of the biomaterial matrix. More water will be immediately absorbed by porous nanofibers with pore size increasing, because the water absorption process is faster than drug movement [143]. Eventually, the drug release rate is increased. Drug molecules could also diffuse out from polymeric nanofibers because of their permeability and thickness [144]. In nondegradable matrix, diffusion is the major factor for drug release. Osmotic pumping is another approach to transport drug molecules via liquid-filled pores. Osmotic pressure drives the influx of liquid; as a result, drug molecules are diffused into surroundings by the force [145,146].

### 6.2. Nanofiber Erosion

The erosion of nanofibers can be divided into surface erosion and bulk erosion. Surface erosion means that the polymeric nanofibers degrade from the surface by slowly reducing the size from the outside towards the inside [147]. Surface erosion is an ideal drug release mechanism for drug delivery application, due to the erosion kinetics being controllable as well as reproducible. It is notable that surface erosion occurs when the polymeric nanofiber erosion rate is higher than the liquid penetration rate into the bulk nanofibers [148]. Besides, bulk erosion occurs when the liquid penetration rate is higher than the polymeric nanofiber erosion rate. Therefore, bulk erosion is similar to hydrolyzing the polymeric nanofibers. Compared with surface erosion, bulk erosion is hard to control and the drug molecules cannot be protected from the environment [147].

### 6.3. Drug Release Profile

The models of various drug release profiles are shown in Figure 12. A zero-order release model is an ideal profile for drug release-rate-controlling. It means that the drug release rate is constant at any moment during the period of release. A controlled drug release system can not only provide temporal and spatial control of drug release, but also protect the drugs or bioactive molecules with therapeutic efficiency. Designed drug delivery formulations can be used to improve drug efficiency so as to minimize drug dose in patients.

However, the most common drug release is a tri-phasic profile, and it sometimes represents a bi-phasic [142]. In a classical tri-phasic, phase I is often described as a burst release, which usually involves non-capsulation of drug molecules on the surface of the nanofibers; phase II usually shows a slow release, which is dominated by slow drug diffusion; phase III will be a faster release because of nanofiber erosion, compared with phase II. For example, PLGA-based nanofiber is an excellent biodegradable polymer material with a typical tri-phasic profile [142]. In PLGA micro/nano systems, polymer concentration and the morphology of the nanofibers are the key parameters in controlling drug release in phase I and II, and the drug release in phase III is determined by the degradation rate [149]. The drug release profiles of polymeric nanofibers are dependent on their individual physicochemical properties; therefore, the drug release profile needs to be studied for each polymeric matrix [141].

## 7. Conclusions and Future Prospects

Functional polymeric nanofibrous biomaterials are a critical class of artificial ECM for drug delivery in biomedical applications. Biomaterials from different polymers have been employed to load various drugs and produce multi-functional nanofibers. In order to control drug release rate while maintaining the bioactivity of agents, many approaches to adsorbing drugs have been investigated. Multiaxial nanofibers might be the best method for loading multiple drugs and delivering them with controllable release, because different drugs can be encapsulated into different layers to satisfy the multi-functionalization of nanofibers. The development of nanofiber production technology also has improved dramatically, and various methods are employed to fabricate multiaxial nanofibers with multiple functions for drug delivery. However, no technology has been used to produce multi-functional nanofibers for drug delivery at an industrial scale.

In the future, the mass, stable, and efficient production of multi-functional nanofibrous biomaterials should be the main the focus of technology development to translate the advanced nanofibers from the laboratories to industry implementation and eventually to clinical application. In addition, further studies should focus on combining nanofiber production and 3D printing technology. Therefore, the produced nanofibers can mimic both the micro and macro structures of different native ECMs, so as to apply functional nanofibers into individual tissues effectively and improve the functions of nanofibers in biomedical applications.

## Figures and Tables

**Figure 1 pharmaceutics-12-00522-f001:**
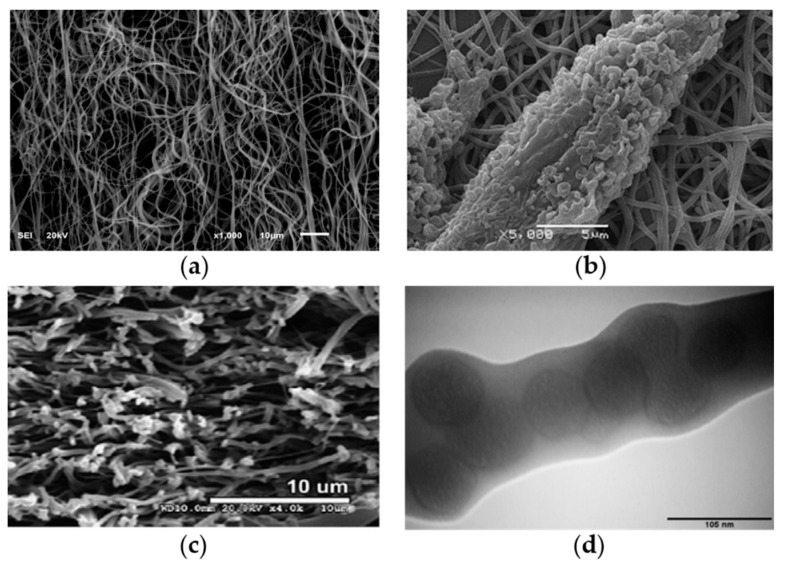
SEM images of different composite nanofibers for various biomedical applications: (**a**) chitosan–polyethylene oxide (PEO) composite nanofibers [43] (Copyright 2019, MDPI); (**b**) L929 cell seeded on carboxyethyl chitosan/polyvinyl alcohol (PVA) nanofibrous membrane after 48-h culture [44] (Copyright 2011, Elsevier); (**c**) cross-section of PCL/collagen nanofiber scaffolds [45] (Copyright 2009, Elsevier); (**d**) TEM image of highly aligned poly lactic-*co*-glycolic acid (PLGA)–gelatin nanofibers with 10 wt.% mesoporous silica nanoparticles [33] (Copyright 2015, Elsevier).

**Figure 2 pharmaceutics-12-00522-f002:**
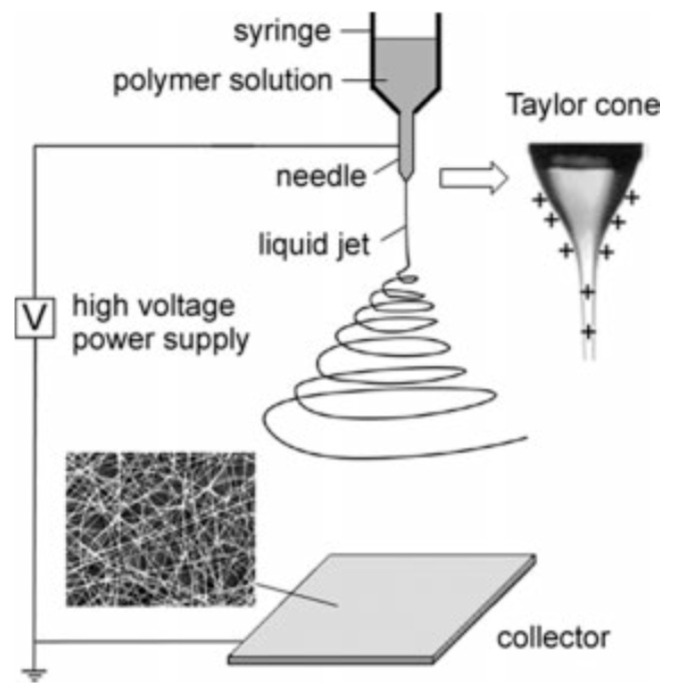
Schematic diagram of the basic setup for electrospinning process and Taylor cone [98] (Copyright 2004, John Wiley and Sons).

**Figure 3 pharmaceutics-12-00522-f003:**
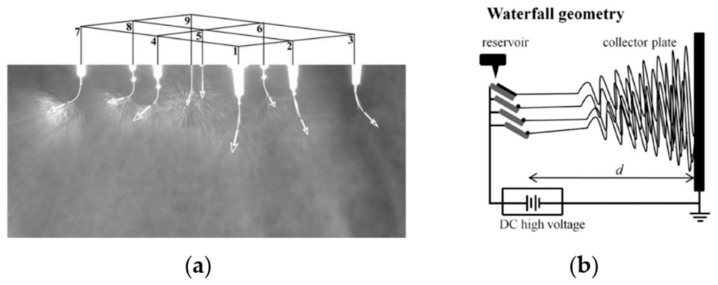
(**a**) Photograph of multi-jet electrospinning [99] (Copyright 2005, Elsevier); (**b**) schematic diagram of waterfall geometry electrospinning setup [100] (Copyright 2010, Elsevier).

**Figure 4 pharmaceutics-12-00522-f004:**
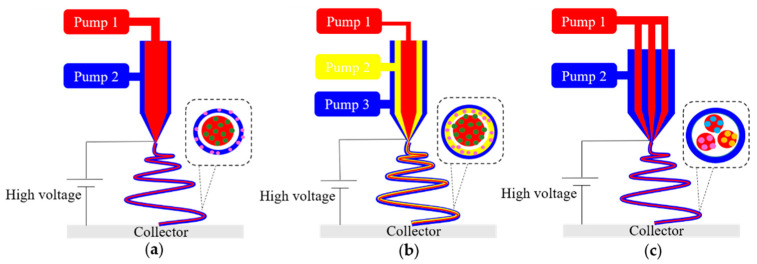
Schematic drawings of multiaxial electrospinning device for fabricating various components for encapsulating multiple drugs. (**a**) Coaxial electrospinning, (**b**) triaxial electrospinning, (**c**) islands-in-the-sea electrospinning.

**Figure 5 pharmaceutics-12-00522-f005:**
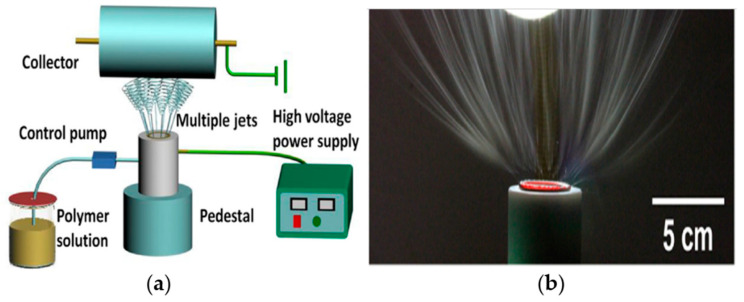
(**a**) Schematic diagram of a needleless electrospinning system, and (**b**) a photograph of the multiple jets forming in the needleless electrospinning process [103] (Copyright 2019, Elsevier).

**Figure 6 pharmaceutics-12-00522-f006:**
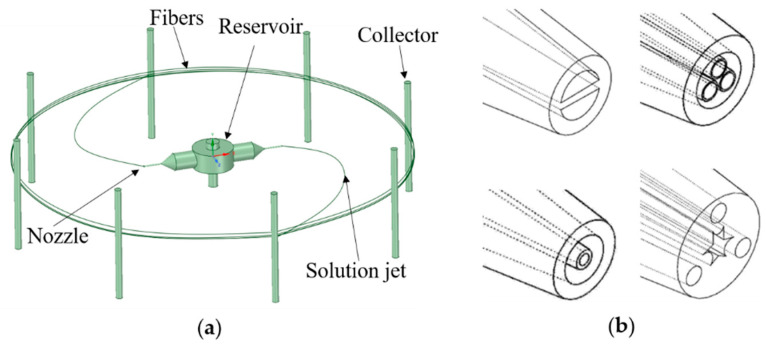
(**a**) Schematic of the centrifugal spinning process; (**b**) various cross-section shapes of nozzles for diverse morphologies of nanofibers in centrifugal spinning [106] (Copyright 2015, China Intellectual Property Office).

**Figure 7 pharmaceutics-12-00522-f007:**
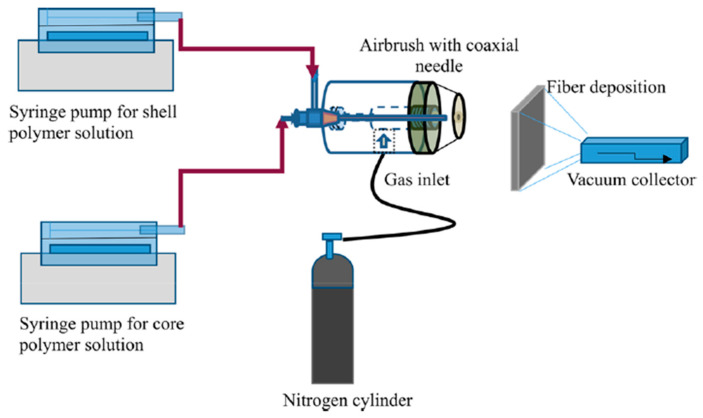
Schematic diagram of solution blowing spinning setup for single and core–shell structural nanofibers [109] (Copyright 2018, American Chemical Society).

**Figure 8 pharmaceutics-12-00522-f008:**
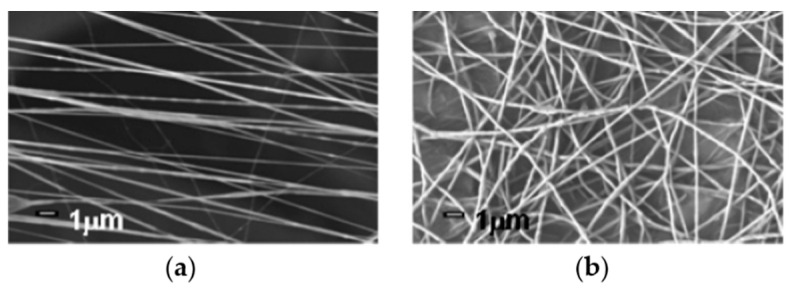
(**a**) SEM images of isotropic nanofibers and (**b**) anisotropic nanofibers [115] (Copyright 2007, American Chemical Society).

**Figure 9 pharmaceutics-12-00522-f009:**
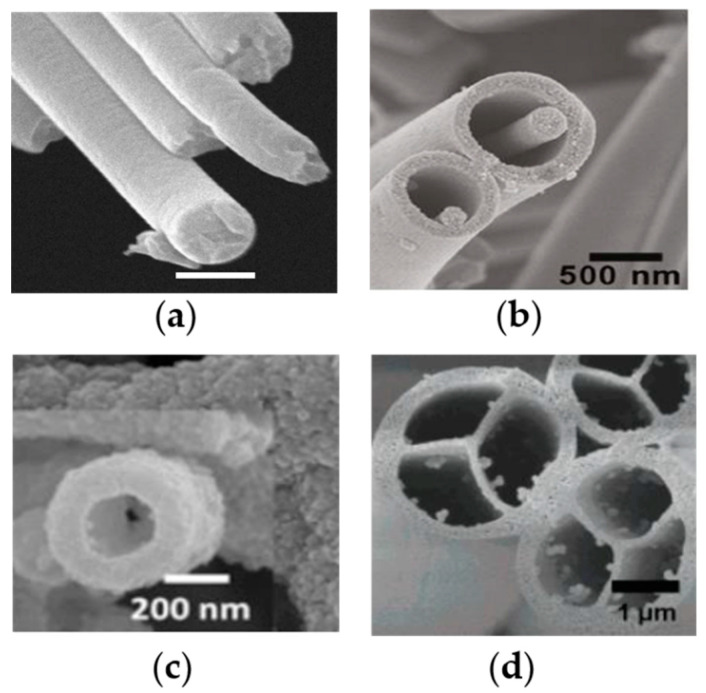
SEM images of various structural nanofibers: (**a**) single nanofibers, (**b**) core–shell nanofibers [125] (Copyright 2010, American Chemical Society); (**c**) hollow nanofibers [126] (Copyright 2017, Elsevier) and (**d**) islands-in-the-sea nanofibers [124] (Copyright 2007, American Chemical Society).

**Figure 10 pharmaceutics-12-00522-f010:**
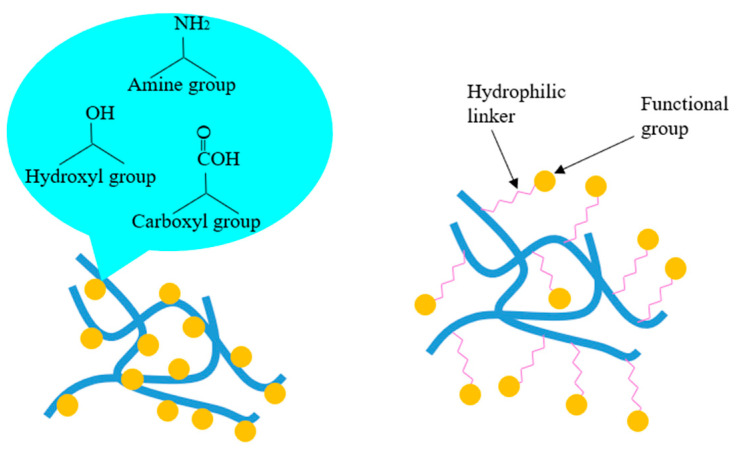
Common chemical adsorption of drug molecules onto the surface of nanofibers.

**Figure 11 pharmaceutics-12-00522-f011:**
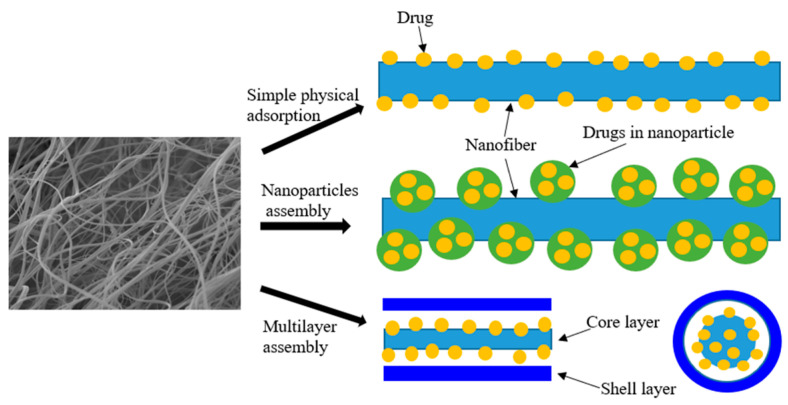
Three different physical adsorptions of drugs into nanofibers.

**Figure 12 pharmaceutics-12-00522-f012:**
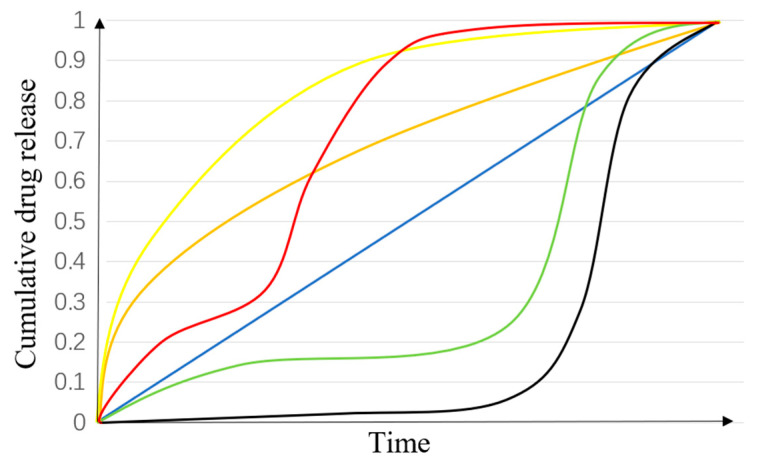
Different drug release profiles. Red: tri-phasic release with a short phase II; yellow: burst release and a quick phase II; blue: a zero-order release; orange: burst release with a zero-order release; green: a classical tri-phasic release; black: a classical bi-phasic, which is similar to a tri-phase without the burst release.

**Table 1 pharmaceutics-12-00522-t001:** Fibers from different polymeric biomaterials for biomedical applications.

Polymer(s)	Solvent	Concentration	Applications	Ref.
Collagen-PCL	HFIP	6 w/v%	Tissue engineering	[18]
Gelatin-PCL	HFIP	6 w/v%	Tissue engineering	
PEO	Colloidal silica	6–10 wt.%	Biosensors	[19]
Collagen-PEO	Hydrochloric acid	1 wt.%	Wound healing, tissue engineering, and hemostatic agents	[20]
		1–2 wt.%		[21]
Silk-PEO	HFIP	4.8–8.8 w/v%	Biomaterial scaffolds	[22]
Chitosan-PEO	Acetic acid	2.5 w/v%	Wound healing	[23]
PLA	DMF	4–9 wt.%	Tissue engineering	[24]
PMMA-SWCNTs	N/A	2–5 wt.%	Biosensor	[25]
PAM	Colloidal silica	6–10 wt.%	Biosensors	[19]
PLLA-PLGA	THF/DMF	1–15 wt.%	Tissue engineering	[26]
PLGA-collagen	THF/DMF	20 wt.%	Bioengineered skin	[27]
PLGA	HFIP	24 w/v%	Tissue engineering	[28]
	HFIP	5wt.%	Peripheral nerve regeneration	[29]
	Chloroform	5wt.%	Tissue engineering	[30]
	THF/DMF	10 w/v%	Bone tissue engineering	[31]
PLGA-gelatin-MSNPs	HFIP	17 wt.%	Cell culture and tissue engineering	[32,33]
Dox-MWCNTs-PLGA	THF/DMF	1–3 wt.%	Drug delivery	[34]
Chitosan-PEG-PLGA	Ethyl acetate	18 wt.%	Tissue engineering	[35]
PLGA-PU	DCM or HFIP	50 w/v%	Drug delivery	[36]
PLGA-SF-CL	HFIP	5wt.%	Peripheral nerve regeneration	[29]
PCL-PLGA	HFIP	15 w/v%	Drug delivery	[37]
XN-PLGA	Chloroform	10 wt.%	Tissue engineering	[30]
Cellulose-Chitosan	DMF	1 w/v%	Bone tissue engineering	[38]
Chitosan-PVP	Acetic acid, sodium hydroxide	2.5 w/v%	Wound healing	[39]
Cellulose-TCMC-PEO	Acetone, DMF, chloroform	3 w/v%	Drug delivery	[40]
Cellulose-chitosan-PEO	Sulfuric acid	1.2 w/v%	Tissue engineering scaffolds	[41]
Chitin-chitosan	Sodium hydroxide and acetic acid	0.75 w/v%	Wound dressing	[42]

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
