# Peer review of "Functional Nanofibrous Biomaterials of Tailored Structures for Drug Delivery—A Critical Review"

_pharmaceutics, 2020, doi:10.3390/pharmaceutics12060522_

Round 1

Reviewer 1 Report

The manuscript " Functional nanofibrous biomaterials for controlled drug delivery – a critical review " authored by ZHEN LI and co-workers is a comprehensive review article, overviewing thoroughly and critically several types of nanofibers, approaching drug delivery. In my opinion is an important contribution in the area of novel biomaterials for biomedicine applications. Only few minor issues should be addressed. I have listed briefly here: although is already good, English language needs a bit to be polished by native speaker, literature references are adeguate but maybe reducible on a more concise "fashion". Copyright acknowledgments should be probably refered in figure captions (please check it). Figures are Ok. Only one doubt about the title: the word "controlled" is probably too "challenging" since within the review this property is not really so deeply addressed, maybe something like "tailored" finds a better place.   

Reviewer 2 Report

The manuscript by Li et al, entitled “Functional nanofibrous biomaterials…” provides a comprehensive review of the nanofibrous biomaterials fabrication by electrospinning, centrifugal spinning and solution blowing methods using a series of natural and synthetic biomaterials for drug delivery and tissue culture applications. The authors have covered the use of most common biomaterials such as collagen, chitosan, PEO, PLGA, and PCL.

A few comments:

This review will be a very useful reference for the graduate students and postdocs who are getting into this field. The authors have covered the nanofiber fabrication methods, suitable polymers and drug delivery applications.

There are several reviews available on nanofiber fabrication by electrospinning methods.

The drug loading and drug release from the nanofiber sections are weak and superficially discussed. Need more in depth review
